# CycleVTON: Improving Diffusion-Based Virtual Try-On with Cycle-Consistent Training

## Abstract

We present CycleVTON, a cycle-consistent diffusion-based virtual try-on framework. Unlike existing methods that rely on a single try-on network, our model consists of two conjugated networks. In addition to the regular try-on network, we design a clothing extraction network that extracts the clothing worn by the person and standardizes it into a front-facing format. These two networks are symmetrical, enabling alignment between the generated dressed human and real images of dressed human, as well as between the extracted clothing and its front-facing ground truth. This cycle-consistent optimization strategy allows for enhanced retention of clothing textures and structures, ensuring a more realistic and accurate clothing generation in virtual try-on scenarios. Moreover, the conjugated network structure not only supports traditional virtual try-on but also allows flexible clothing extraction and clothing exchange between different individuals. The experiments on VITON-HD demonstrate the effectiveness of our approach.

## 1 Introduction

In recent years, with the rapid development of online shopping, virtual try-on technology has garnered significant attention as an important technological application. The goal of this technology is to realistically synthesize given clothing images onto the corresponding regions of a person's body. This task poses significant challenges due to the need to maintain the individual characteristics of the person, retain the style of the clothing along with its details, and ensure that the virtual try-on appears natural and well-fitted.

Early virtual try-on systems (Han et al., 2018; Wang et al., 2018) primarily relied on Thin-Plate Spline (TPS) transformations to achieve clothing deformation. While this approach could roughly fit clothings to the human body and worked for simple poses, it struggled to handle more complex human poses. To more accurately predict clothing deformation, subsequent research (Xie et al., 2023) introduced appearance flow estimation to model non-rigid deformations, addressing complex poses by predicting dense pixel-level correspondences. Although such warping-based virtual try-on methods have achieved impressive results, particularly in retaining pattern details due to pixel-level predictions, they often produce unnatural results and artifacts when dealing with occlusions and intricate human poses, see Figure 1 (a).

Due to remarkable capabilities in visual generation demonstrated by diffusion models (Ho et al., 2020; Song et al., 2020; Rombach et al., 2022; Ramesh et al., 2022), they have been increasingly applied to virtual try-on tasks. Compared to warping-based methods, methods (Gou et al., 2023; Morelli et al., 2023; Zhu et al., 2023; Kim et al., 2024; Chen et al., 2024; Xu et al., 2024; Choi et al., 2024; Zhang et al., 2024) based on diffusion models can handle complex human poses, achieving more natural try-on effects. However, diffusion models generate the dressed person progressively from noise, and their supervision signal is limited to the original clothing, rather than using the deformed clothing. Instead, the model implicitly transforms the original clothing, enabling an adaptive modification. This often results in the loss of fine details, as shown in Figure 1(b). While many efforts (Xu et al., 2024; Choi et al., 2024) have sought to preserve clothing details, challenges like texture duplication and layout alterations persist, particularly with intricate patterns. Consequently, maintaining the structural layout and texture fidelity of clothing remains a critical challenge.

In this work, we observe that virtual try-on based on diffusion models performs better when the clothing is flat or presented in a near-frontal view, accurately restoring the clothing's detailed texture

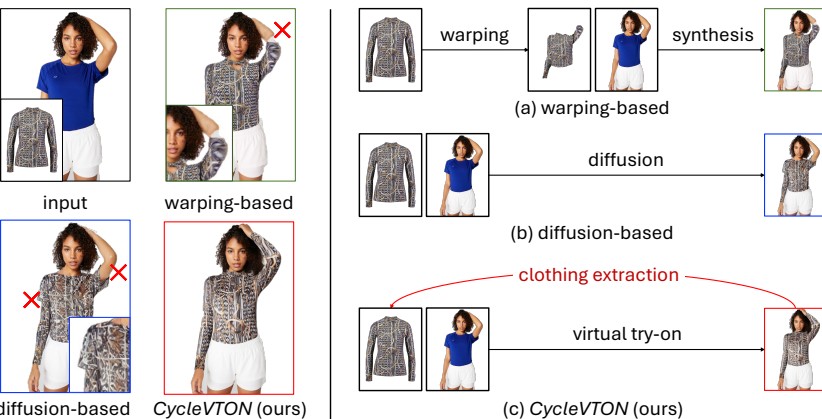

Figure 1: Different methods and try-on effects. Traditional warping-based synthesis faces obvious artifacts. The dressed clothes based on diffusion method might show artifacts, too, while our proposed method can provide a more accurate and vivid try-on result with an additional clothing extraction module in the network design.

and overall appearance. However, when the clothing undergoes significant deformation or changes in angle after being worn, issues such as wrinkling and stretching arise. Even with real images of worn clothing for supervision, it remains challenging to reproduce the original texture and patterns of the clothing accurately.

Based on this point, we propose a virtual try-on network architecture called CycleVTON, as shown in Figure 1(c), which consists of two complementary modules: (1) a virtual try-on network that renders clothes from a standard frontal view onto the corresponding regions of the human body; (2) a clothing extraction network that extracts clothing worn on the body and standardizes it to a frontal view. To jointly optimize these two models, we introduce a cycle consistency optimization strategy, enabling the networks to complement each other. This approach not only uses ground-truth images of worn clothing for supervision but also learns from clear, frontal-view clothing, thereby preserving the overall appearance and fine details of the clothing.

Our model offers several advantages over previous methods: (1) *Improved accuracy*: enabling a better representation of the overall appearance and specific details of clothing, with improved adaptability to clothing deformation and human movements; (2) *Enhanced flexibility*: as it combines virtual try-on with clothing take-off networks, allowing not only traditional virtual try-on but also clothing extraction and clothing swapping between different individuals; (3) *Simplified supervision data acquisition*, as it is no longer limited to front-view clothing input and can directly utilize any clothing worn on a person for input and training.

In summary, our contributions are as follows:

(1) We propose a new cycle-consistent diffusion-based framework for virtual try-on, named CycleVTON. This framework not only enhances the accuracy of clothing generation and improves robustness to clothing deformation and human pose variation but also provides greater flexibility to meet diverse clothing generation needs.

(2) We introduce two complementary networks in CycleVTON. Alongside the conventional try-on network, we innovatively incorporate a clothing extraction network that extracts clothing from the human body and normalizes it to a frontal view. We propose a cycle-consistency optimization strategy, which aligns the generated dressed human with real images and the extracted clothing with its true frontal appearance. This approach expands the model's supervision signals and enhances its performance.

(3) We conduct extensive experiments between the proposed method and state-of-the-art virtual try-on approaches. The results demonstrate that CycleVTON achieves superior performance on the widely used VITON-HD benchmark. Furthermore, we showcase the model's capability to swap clothing between different individuals through the cycle design.

## 2 RELATED WORK

### 2.1 WARPING-BASED VIRTUAL TRY-ON

arping-based virtual try-on methods typically involve two main stages: clothing warping and try-on synthesis. Early approaches, such as those introduced in (Han et al., 2018), utilized Thin Plate Spline (TPS) for deforming clothing to fit the human body. This method employed a coarse-to-fine strategy, initially generating a rough try-on result with an encoder-decoder architecture, which was then refined using TPS to accurately map the clothing onto the target person. Subsequent methods have focused either on enhancing TPS's deformation capabilities (Fincato et al., 2021; 2022; Li et al., 2023) or addressing the limitations within the try-on workflow (Wang et al., 2018; Yang et al., 2020; Ge et al., 2021a). To improve TPS performance, Minar et al. (2020) redesigned the process with optimized input representations and refined training loss. Similarly, Wang et al. (2018) refined the TPS training process by introducing a new Geometric Matching Module (GMM), moving away from calculating key point correspondences and instead matching in-store clothing to the target body shape. Other approaches have sought to address specific challenges, such as misalignment and occlusion. Ge et al. (2021a) introduced a disentangled cycle-consistent try-on network to mitigate misalignment between clothing and the body, avoiding the need for paired clothing-body data. For occlusion issues, Yang et al. (2020) proposed predicting the semantic layout of the reference image post-try-on to identify which parts should be retained.

However, the limited deformation capabilities of TPS make it challenging to handle complex poses. To address this, many optical flow-based virtual try-on methods (Bai et al., 2022; Ge et al., 2021b; Han et al., 2019; He et al., 2022; Shim et al., 2024) have been proposed, which calculate pixel-level deformation fields for clothing, enabling them to handle complex deformations effectively. Despite these advancements, these methods still suffer from misalignment caused by explicit deformations, often producing unnatural visual try-on results when dealing with complicated human poses. Additionally, explicit deformation fails to account for real occlusion during the try-on process, leading to artifacts like clothing textures incorrectly appearing behind the neck.

### 2.2 DIFFUSION-BASED VIRTUAL TRY-ON

With the remarkable success of diffusion models in visual generation (Ho et al., 2020; Song et al., 2020; Rombach et al., 2022; Ramesh et al., 2022), their application in virtual try-on has garnered increasing research attention. Models like LaDI-VTON (Morelli et al., 2023) and DCI-VTON (Gou et al., 2023) leverage diffusion models to virtually try on clothing that has been explicitly deformed to fit the human body. TryonDiffusion (Zhu et al., 2023), utilizing two parallel UNet networks, demonstrated on large-scale datasets that diffusion models can implicitly warp clothing foregrounds for virtual try-on. OOTDiffusion (Xu et al., 2024) and IDM-VTON (Choi et al., 2024) fine-tune and denoise clothing networks with the same UNet structure to preserve more accurate clothing details. MMTryon (Zhang et al., 2024) combines diffusion models in virtual try-on to consider more modality information and clothing details. Despite these advancements, diffusion model-based approaches still face challenges in preserving fine clothing details and largely depend on front-view clothing. In this work, we revisit the diffusion-based framework for virtual try-on, aiming to enhance detail preservation through cyclically extracting features from the generated results.

## 3 CYCLEVTON

This section begins with an overview of the diffusion model framework that underpins CycleVTON. We then introduce the overall architecture of our model, followed by the details of the virtual try-on network and the clothing extraction network. Lastly, we delve into the design and implementation of the cycle-consistent optimization strategy.

### 3.1 PRELIMINARY

**Stable Diffusion** (Rombach et al., 2022) is a prominent type of latent diffusion model (LDM) designed to efficiently generate high-quality images in a compressed latent space. This architecture employs a variational autoencoder (VAE), which comprises both an encoder and a decoder. The

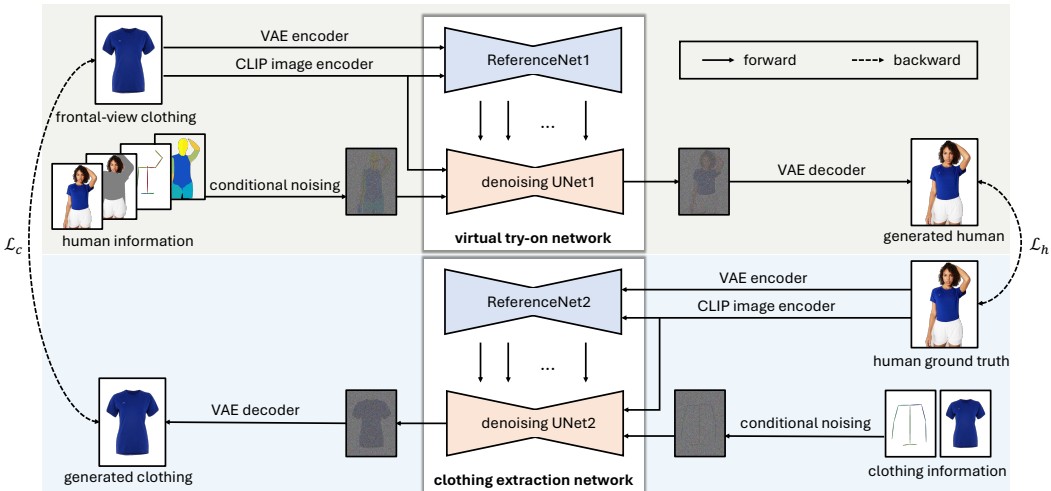

Figure 2: The framework of our proposed CycleVTON, which consists of two main parts: virtual try-on network (VTON) and clothing extraction network (CEN). The extracted clothing through CEN can be further utilized in the try-on module to construct a cycle-consistent training pipeline.

encoder transforms the input image $\mathbf{x}$ into a latent representation $\mathbf{z}$, while the decoder reconstructs the image from this latent space.

During the training phase, a controlled amount of noise is systematically introduced to the latent representation $\mathbf{z}$ over a series of timesteps, resulting in noisy latent states $\mathbf{z}_t$. A UNet architecture is then utilized to learn the denoising process, enabling it to predict and remove the noise, thereby recovering the original latent representation from its perturbed forms.

The training objective is formulated by minimizing the following loss function:

$$\mathcal{L}_{\text{LDM}} = \mathbb{E}_{\mathbf{z}_t, c, \epsilon, t} \left[ \| \epsilon - \epsilon_\theta(\mathbf{z}_t, c, t) \|_2^2 \right],$$

where $\epsilon_\theta$ denotes the denoising function of the UNet, $\epsilon$ represents the Gaussian noise added at each timestep $t$, and $c$ indicates the conditional embeddings, such as text prompts encoded via the CLIP model.

During inference, the process begins with sampling a latent variable $\mathbf{z}_T$ from a Gaussian distribution at the initial timestep $T$. This latent variable undergoes a deterministic denoising procedure, progressively refining it to $\mathbf{z}_0$. The UNet predicts the noise at each step, and after the final denoising operation, the latent representation $\mathbf{z}_0$ is reconstructed into an image via the VAE decoder.

**ReferenceNet** is a feature extraction network introduced by Hu (2024), capable of capturing detailed features from a reference image and integrating them into the Stable Diffusion through spatial attention to maintain consistency in the character's appearance.

**Denoising UNet** is a variant of Stable Diffusion introduced by Hu (2024), which is responsible for generating visual content by progressively removing noise from noise inputs under the guidance of human pose and reference image features.

## 3.2 OVERALL ARCHITECTURE

Our work aims to optimize the performance of diffusion-based virtual try-on framework, improving the retention of texture and structure in virtual try-on systems. To achieve this, we propose a new framework called CycleTryon. Figure 2 illustrates the overall architecture of the framework, which consists of two main modules: the virtual try-on network and the clothing extraction network.

The virtual try-on network uses standard frontal-view clothing as input, rendering it onto the corresponding regions of the human body. The clothing extraction network, on the other hand, extracts

clothing data from images of dressed humans and standardizes it into a frontal-view format. These two networks complement each other: the virtual try-on network can use the output of the clothing extraction network as input, and vice versa, forming a cycle structure that flexibly supports various clothing-related applications. Leveraging this cyclic nature, we introduce a cycle-consistent optimization strategy. This strategy not only allows for supervision using dressed human images, as in traditional virtual try-on models, but also enables the use of frontal clothing images for supervision. This enhances the ability of both networks to retain information from the original clothing and human images, thereby improving the overall performance of the model.

### 3.3 VIRTUAL TRY-ON NETWORK

As illustrated in Figure 2, the virtual try-on network aims to integrate clothing information into the masked clothing area of the human body image while preserving the original body pose and image background. This network comprises two main components: ReferenceNet1 and denoising UNet1. The ReferenceNet1 takes standard frontal-view clothing images as input, combined with CLIP features, to provide more refined clothing details to the denoising UNet1, thereby enhancing the preservation of these details. The denoising UNet1 primarily receives conditional noise containing human information as input and uses both the CLIP features and the clothing features from ReferenceNet1 to progressively denoise and incorporate the clothing details into the corresponding regions of the human body.

For the CLIP features used as conditions, we employ CLIP's image encoder to convert the frontal-view clothing images into image embeddings. These embeddings are then injected into the diffusion model through cross-attention to provide color and texture information.

For the conditional noise, we generate it by combining the human image, the masked human body image, the body pose image obtained via OpenPose (Cao et al., 2017), and the human parsing image obtained using Densepose (Güler et al., 2018). Specifically, the human image, masked human body image, and human parsing image are encoded using a Variational Autoencoder (VAE) (Kingma, 2013) to generate latent variables. The latent variable from the human image is further perturbed with noise to create latent noise. These latent variables are then concatenated with the mask image of the clothing area, which is resized through interpolation, forming a 13-channel tensor. Here, each latent variable has 4 channels, and the mask image has 1 channel. The OpenPose image is processed using PoseGuider (Hu, 2024) to extract 4-channel human body pose features. Finally, the 13-channel tensor is passed through a convolutional layer and then added to the body pose features, producing the final conditional noise with 4 channels.

### 3.4 CLOTHING EXTRACTION NETWORK

The goal of our clothing extraction network is to capture clothing information from human images, thereby creating a standardized clothing representation across various virtual try-on tasks, including traditional cloth-to-human try-on and more flexible human-to-human try-on. This network also includes both ReferenceNet2 and denoising UNet2. However, unlike the virtual try-on network, ReferenceNet2 uses dressed human images as input, providing denoising UNet2 with more accurate and detailed information about distorted clothing. Denoising UNet2 takes conditional noise containing frontal-view posture information as input, leveraging CLIP features with distorted clothing information as a condition. It then progressively removes the noise to extract the frontal-view clothing details from the distorted clothing.

For the CLIP features used as conditions, we utilize CLIP's image encoder to transform the distorted clothing image into an image embedding. For the conditional noise, we generate it by combining the frontal-view clothing image with the clothing pose image. Specifically, the frontal-view clothing image is first encoded using a VAE to obtain latent variables, which are further perturbed to generate latent noise. At the same time, the clothing pose image is processed by PoseGuider (Hu, 2024) to extract pose features. The latent noise is then passed through a convolutional layer and added to the clothing pose features to produce the final conditional noise.

For clothing pose estimation, the focus is capturing key point information specific to the clothing. To obtain training data for clothing pose, we utilize existing human pose estimation methods (such as (Cao et al., 2017)) to predict ten key points on the body that correspond to the upper clothing.

These key points include the neck, two shoulders, two elbows, two wrists, and four hip points. By extracting the positions of these key points from human images, we can derive the pose information of the clothing worn. Next, we segment the clothing region from the human body to generate an image corresponding to the identified clothing pose. We then pair the clothing image with its associated pose and use an existing clothing landmark predictor (Chen et al., 2023) for training. By providing the predictor with normal clothing images, we can effectively acquire the corresponding pose information.

By introducing the clothing extraction network, our model can not only utilize the paired clothing-human images provided in the dataset but also extract additional clothing images from humans, thereby enhancing the training of the virtual try-on network. This approach provides extra supervised training, significantly improving the model's performance in terms of accuracy, robustness, and generalization. These improvements enable the virtual try-on network to achieve higher accuracy and better clothing fitting, resulting in more effective and precise clothing rendering.

### 3.5 CYCLE-CONSISTENT OPTIMIZATION STRATEGY

To enhance the quality of generated clothing and maintain consistency before and after distortion, we introduce a cycle-consistent optimization strategy. As shown in Figure 2, is composed of two components: the human rendering loss $\mathcal{L}_h$ and the clothing rendering loss $\mathcal{L}_c$. The formulation can be written as:

$$\mathcal{L} = \mathcal{L}_h + \mathcal{L}_c. \tag{1}$$

The calculation of $\mathcal{L}h$ involves two scenarios: (1) *Regular dressed human generation*: In this case, the ground truth of the front-view clothing is used as the input to ReferenceNet1. After obtaining the guiding conditions, denoising UNet1 generates the dressed human. The diffusion loss is then computed using the loss $\mathcal{L}_{\text{LDM}}$ and the ground truth of the dressed human. which we denote as $\mathcal{L}_h^{\text{regular}}$. This scenario is only used to train the virtual try-on network. (2) *Cyclic dressed human generation*: Here, the front-view clothing features extracted by the clothing extraction network are used as the input to ReferenceNet1. Similarly, denoising UNet1 generates the dressed person, and the diffusion loss is computed, marked as $\mathcal{L}_h^{\text{cyclic}}$. In this case, both the virtual try-on network and the clothing extraction network are trained simultaneously. Consequently, $\mathcal{L}_h$ is formulated as:

$$\mathcal{L}_h = \mathcal{L}_h^{\text{regular}} + \omega \mathcal{L}_h^{\text{cyclic}}. \tag{2}$$

Here, $\omega$ is a binary value (0 or 1) that controls whether $\mathcal{L}_h^{\text{cyclic}}$ is included in the calculation. In the diffusion model, we derive the clothing image features $z_0$ from the latent noise $z_t$ using the following equation:

$$z_0 = \left(z_t - \sqrt{1 - \alpha_t}\epsilon\right)/\sqrt{\alpha_t}. \tag{3}$$

Here, $\{\alpha_t\}_{t=1}^T$ is a noise schedules, $\epsilon \sim \mathcal{N}(0, 1)$ is a Gaussian noise. $\omega$ is set to 0 when $t$ exceeds a certain threshold; otherwise, it is set to 1 when $t$ is below or equal to this threshold. As $t$ increases, $z_t$ becomes noisier and loses details from the original image, resulting in less precise image features. This imprecision hinders the joint optimization of our model. Therefore, we define a threshold for $t$ to turn off $\mathcal{L}_h^{\text{cyclic}}$ when the noise level is too high.

For the computation of $\mathcal{L}_c$, similarly, we also divide it two scenarios: regular clothing generation and cyclic clothing generation, which allows us to write:

$$\mathcal{L}_c = \mathcal{L}_c^{\text{regular}} + \omega \mathcal{L}_c^{\text{cyclic}}. \tag{4}$$

In this way, this strategy implements joint training of the virtual try-on network and the clothing extraction network, making full use of the real annotated data of dressed humans and frontal-view clothing for synchronized supervision. Additionally, it imposes consistency constraints on the outputs of both networks, ensuring that structural and texture detail information is preserved during the try-on process.

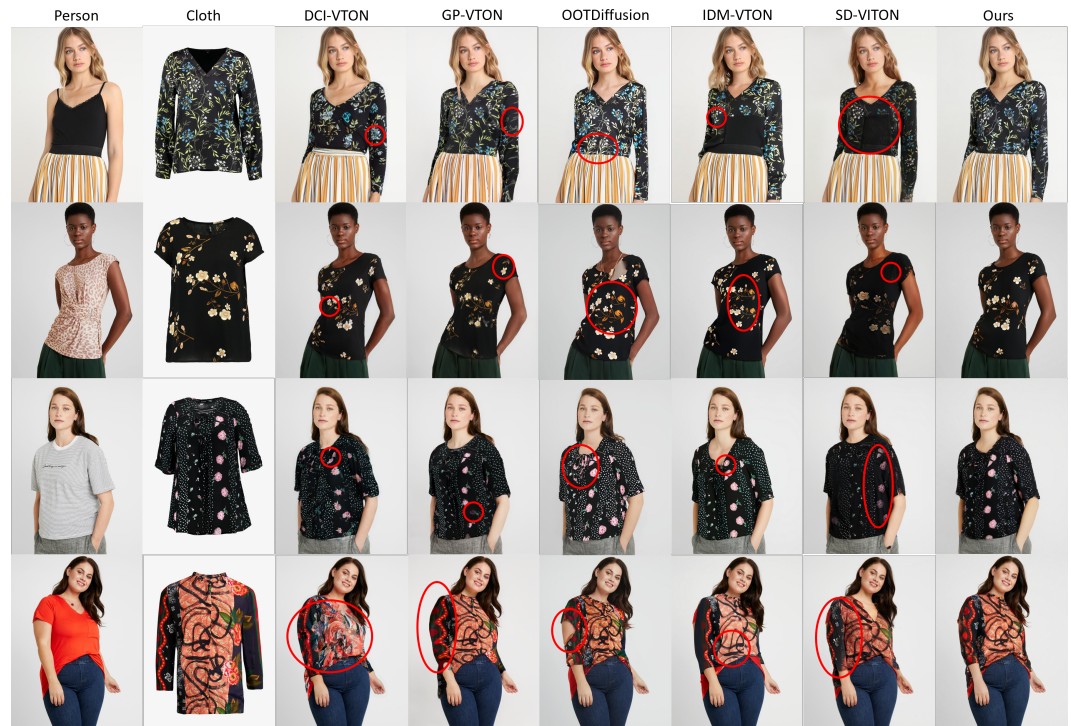

Figure 3: Qualitative comparison on the VITON-HD dataset. The red circles show artifacts, inconsistent textures, and unnatural effects.

## 4 EXPERIMENTS

### 4.1 EXPERIMENTS SETTING

**Dataset:** Our experiments primarily utilize the publicly available virtual try-on dataset, VITON-HD (Choi et al., 2021), which comprises high-resolution in-shop clothing images that include both paired and unpaired model images. Derived from VITON, the VITON-HD dataset comprises 13,679 pairs of frontal female and upper-body clothing images, divided into training and testing sets with 11,647 and 2,032 pairs, respectively.

**Evaluation Metrics:** Similar to previous works (Choi et al., 2021), virtual try-on involves paired and unpaired settings. In the paired setting, the original clothing image is used to reconstruct the original person image, while in the unpaired setting, the person's clothing is changed to the given clothing. To assess the performance of our method, we employ distinct evaluation metrics for each setting, as indicated in the existing literature. For the paired setting, we utilize two widely-used metrics: Structural Similarity (SSIM) (Wang et al., 2004) and Learned Perceptual Image Patch Similarity (LPIPS) (Zhang et al., 2018). In contrast, for the unpaired setting, we evaluate our model using Frechet Inception Distance (FID) (Heusel et al., 2017) and Kernel Inception Distance (KID) (Bińkowski et al., 2018).

### 4.2 EXPERIMENTAL RESULTS

We compare the performances of the proposed CycleVTON (Ours) with several state-of-the-art virtual try-on methods on the VION-HD test set. These methods include warping-based approaches which usually use optical flow for transformation such as GP-VTON (Xie et al., 2023) and SD-VITON (Shim et al., 2024), as well as diffusion model-based approaches such as DCI-VTON (Gou et al., 2023), LaDI-VTON (Morelli et al., 2023), IDM-VTON (Choi et al., 2024), and OOTDiffusion (Xu et al., 2024). Table 1 presents a quantitative result, from which we can see that warping-based methods have an advantage in SSIM and LPIPS metrics, as their pattern generation tends to be

Table 1: Quantitative comparison with other try-on methods.

| Method | SSIM ↑ | LPIPS ↓ | FID ↓ | KID ↓ |
|---|---|---|---|---|
| DCI-VTON (Gou et al., 2023) | 0.8922 | 0.0720 | **8.7596** | 0.90 |
| GP-VTON (Xie et al., 2023) | 0.8907 | 0.0866 | 9.6158 | 1.40 |
| LaDI-VTON (Morelli et al., 2023) | 0.876 | 0.0910 | 9.4100 | 1.67 |
| IDM-VTON (Choi et al., 2024) | 0.8774 | 0.0809 | 9.1874 | 1.21 |
| OOTDiffusion (Xu et al., 2024) | 0.8542 | 0.0945 | 9.3860 | 1.10 |
| SD-VITON (Shim et al., 2024) | 0.8850 | 0.0917 | 9.7047 | 1.50 |
| CycleVTON (Ours) | **0.8964** | **0.0661** | 9.1447 | **0.90** |

more accurate compared to diffusion models. However, in more complex scenarios, these methods often produce results that are less realistic and natural than those generated by diffusion-based approaches, which leads to inferior performance in FID and KID metrics. Compared to the baselines, our method demonstrates superior performance in both paired and unpaired settings, highlighting the effectiveness and robustness of our network architecture and training strategy.

More specifically, Figure 3 presents a qualitative comparison. It can be observed that warping-based methods have better detail preservation capabilities in the generated patterns compared to diffusion-based models. However, when handling complex poses, the results generated by warping-based methods often appear unnatural and exhibit noticeable artifacts. In contrast, diffusion-based models significantly improve upon this issue. Compared to other diffusion-based methods, our method demonstrates superior performance in generating texture layouts, especially when dealing with complex clothing textures. Previous methods often struggle with texture duplication or layout changes in such scenarios. Our method, on the other hand, produces texture layouts that more closely adhere to the reference clothing. This demonstrates that incorporating the clothing extraction network significantly enhances the virtual try-on network's ability to preserve texture layout, leading to more precise texture placement.

## 4.3 ABLATION STUDY

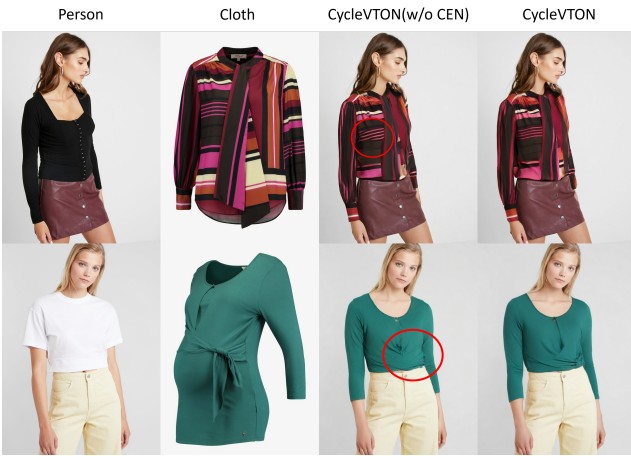

Figure 4: Visualization of try-on results w/ and w/o clothing extraction network (CEN). The red circles show unsatisfied generated details through CycleVTON (w/o CEN), while w/ CEN can achieve better results.

To validate the effectiveness of the proposed modules, we conducted a series of extensive ablation experiments. These experiments aimed to evaluate the contribution of each component within our CycleVTON. However, it is important to note that the configurations used in this study were not based on the optimal settings, which would require higher resolution, resulting in significant time consumption. Instead, we utilized a reduced configuration to facilitate rapid experimentation. Consequently, the data obtained may differ from that derived from the best settings. However, the data

obtained from experiments conducted under paired conditions at relatively low resolutions enable us to quickly and effectively evaluate the effectiveness of each component.

**Effectiveness of the clothing extraction network:** To demonstrate the effectiveness of the clothing extraction network, we investigated its impact on model performance. Specifically, for the model without the clothing extraction network, we independently trained the virtual try-on network using $L_{\text{restruct1}}$. In contrast, for the model incorporating the clothing extraction network, we conducted experiments with and without the cycle-consistency optimization strategy. In the experiment without the cycle-consistency optimization, we first independently trained the clothing extraction network using $L_{\text{restruct2}}$. Subsequently, we froze the parameters of the clothing extraction network and employed $L_{\text{cycle2}}$ to provide additional supervision to the virtual try-on network. The experiment utilizing the cycle-consistency strategy aligns with the methodology outlined in Section 3. The comparison between the first and second rows of Table 2 clearly indicates that employing the clothing extraction network to supervise the virtual try-on network enhances its performance, with all evaluation metrics showing significant improvements. This finding substantiates the efficacy of our clothing extraction network. Figure 4 presents a comparative analysis of our method versus the approach without the clothing extraction network. As illustrated in the upper row, our method more accurately generates striped structures compared to the method lacking the clothing extraction network. In the lower row, even when processing side-view images of clothings, the generated clothes details remain correct, e.g., the bow position.

**Effectiveness of the Cycle-Consistency Optimization Strategy:** The comparison between the second and third rows of Table 2 clearly demonstrates that training with the cycle-consistency optimization strategy yields superior performance, with all evaluation metrics exhibiting substantial improvements. This validates the effectiveness of our cycle-consistency strategy, indicating that it not only facilitates one sub-network supervising another but also enhances each sub-network's ability to retain original image information. Based on the cycle-consistency design, our method can obtain the capability to swap clothing between two different individuals. Figure 5 shows two examples of human-to-human try-on results, which are quite natural and vivid.

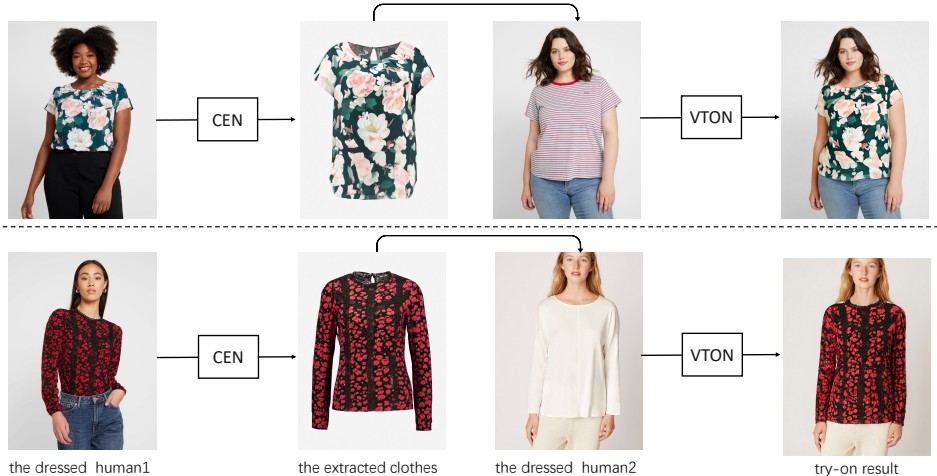

the dressed  human1      the extracted clothes      the dressed  human2      try-on result

Figure 5: The human-to-human try-on results through our proposed CycleVTON. The extracted clothing (from the dressed human1) through the clothing extraction network (CEN) can be tried on by the dressed human2 through the virtual try-on network (VTON).

**The impact of threshold settings in the cycle-consistent optimization strategy:** To evaluate the optimal threshold settings for the cycle-consistency optimization strategy, we conducted experiments with multiple threshold configurations. Specifically, we performed experiments using thresholds of $t = 0$, $t = 200$, $t = 300$, and $t = 400$. The results indicate that as the threshold $t$ increases, the model performance gradually improves, peaking at $t = 300$. Beyond this point, further increases in $t$ result in diminished performance. This decline can be attributed to high-noise images containing less original image information, making it challenging to revert to the original image. Consequently,

Table 2: Ablation study on the effectiveness of clothing extraction network (CEN) and cycle-consistent optimization strategy (CCOS) in CycleVTON.

| Strategy | CEN | CCOS | SSIM ↑ | LPIPS ↓ | FID ↓ | KID ↓ |
|----------|-----|------|--------|---------|-------|-------|
| 1 | × | × | 0.8730 | 0.0720 | 6.5780 | 1.7 |
| 2 | ✓ | × | 0.8741 | 0.0716 | 6.6219 | 1.7 |
| 3 | ✓ | ✓ | 0.8748 | 0.0689 | 6.2214 | 1.3 |

Table 3: Ablation study on the threshold settings of time step $t$ in cycle-consistent optimization strategy.

| Threshold ($t$) | SSIM ↑ | LPIPS ↓ | FID ↓ | KID ↓ |
|-----------------|--------|---------|-------|-------|
| 0 | 0.8730 | 0.0720 | 6.5780 | 1.7 |
| 200 | 0.8748 | 0.0707 | 6.4184 | 1.5 |
| 300 | 0.8748 | 0.0689 | 6.2214 | 1.3 |
| 400 | 0.8747 | 0.0700 | 6.2424 | 1.3 |

this hampers the supervision of the other network and undermines the capability of the sub-network to retain original image information.

## 5 CONCLUSION

In this paper, we present the *CycleVTON* architecture, a significant advancement in virtual try-on technology that effectively addresses the challenges of maintaining the identity of the model while accurately representing clothing details. We design a clothing extraction network that standardizes various clothing conditions into a unified format, and enhances both texture preservation and structural integrity during the try-on process.

Additionally, the cycle-consistent optimization strategy developed for this framework promotes mutual learning between the clothing extraction and virtual try-on networks, allowing for improved retention of the texture and structure in the final synthesized images. Our experiments demonstrate that CycleVTON outperforms existing state-of-the-art methods, as evidenced by superior qualitative and quantitative results on the VITON-HD dataset.

Overall, CycleVTON not only contributes to the ongoing advancements in virtual try-on technology but also sets the stage for further research aimed at refining these methods. Future investigations could explore integrating additional data sources and enhancing the model's adaptability to diverse clothing types and poses, thereby pushing the boundaries of realistic virtual try-on experiences in online shopping.

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

## A  APPENDIX

You may include other additional sections here.

