# OpenReview forum: "CycleVTON: Improving Diffusion-Based Virtual Try-On with Cycle-Consistent Training"
_ICLR.cc/2025/Conference — ICLR 2025 Conference Withdrawn Submission_

### Official Review · Reviewer_zUfa · 2024-10-20

**Soundness:** 3
**Presentation:** 1
**Contribution:** 2
**Rating:** 5
**Confidence:** 3

**Summary:**

The paper proposes a 2D virtual try-on network capable of: 1) rendering an image of a human wearing a given piece of clothing based on an input human image and clothing image, and 2) extracting the image of clothing from a clothed human image. Existing methods have been utilized to extract information from the human image in task 1), such as pose, human parsing, and body mask. A cycle consistency optimization strategy is introduced to jointly optimize the two models for tasks 1) and 2), where the ground truth of model 1) can serve as input for model 2), and vice versa. This approach simplifies supervision and data acquisition while achieving a better representation of the overall appearance.

**Strengths:**

1. The cycle consistency optimization strategy enables the two models to be trained jointly, providing the capability for both virtual try-on and virtual take-off of clothing.

2. The input for the clothing image is not limited to a frontal view; the model can accept variations in the clothing image, which makes it more practical for real-world applications.

**Weaknesses:**

1. The explanation of the methodology could benefit from clearer clarification, as it is difficult to understand which phrase refers to which "image" in the pipeline. For instance, in lines 261-267, adding notations and labeling components in Figure 2 would help avoid ambiguity. Readers may struggle to differentiate between the frontal-view image and the distorted image and add labels (e.g., A, B, C) to each image in Figure 2 and use these labels consistently in the text description will be helpful. Additionally, in Figure 2, using different clothing items for the try-on and take-off procedures would help reduce confusion, as presenting the same blue t-shirt makes the process harder to follow.

2. The experiments were conducted on a dataset with professional fashion images, but in real-world virtual try-on applications, "in the wild" images are typically used. However, there is no experiment demonstrating how the proposed method performs on in-the-wild images. Could this suggest that the model may be overfitting to the fashion dataset and might struggle to adapt to more diverse, real-world images?  the authors could conduct additional experiments on a dataset of non-professional, real-world images to demonstrate the model's generalization capabilities.

3. The poses in the presented dataset appear to involve relatively limited movement. It would be beneficial to include examples or conduct additional experiments with more extreme or dynamic poses to demonstrate the model's robustness.

4. L112 typo: ‘arping’ →’Warping’

**Questions:**

1. L258: Does the term "frontal-view posture information" refer to a canonical pose, or can it be any pose that is frontal? Additionally, is it correct to assume that the generated clothing image will be in this pose?

2. Regarding the clothing extraction model, the input includes a photo of the clothed human, an image of the clothing itself, and the pose information. If the image of the clothing is already available, what is the purpose of using this model to extract the clothing again? Please clarify if there is a misunderstanding in my interpretation.

3.  Is there any limitations to the types of poses the model can handle?

---

### Official Review · Reviewer_gRot · 2024-11-01

**Soundness:** 3
**Presentation:** 3
**Contribution:** 3
**Rating:** 5
**Confidence:** 5

**Summary:**

This paper introduces a cycle-consistent and diffusion-based framework for try-on. It consists of two networks: a regular try-on network and a clothing extraction network. The latter standardizes the clothing into a frontal view, allowing for a more realistic and accurate representation. Its consistent strategy optimizes the alignment between generated try-on images and original human images, as well as between extracted clothing and its accurate frontal view. This framework enables traditional virtual try-ons and adaptable applications such as clothing extraction and swapping. Experiments on the VITON-HD benchmark confirm CycleVTON’s superior performance compared to existing models. However, challenges remain in handling significant deformations and changes in clothing texture or orientation. Furthermore, the cycle-consistent idea is not new, and please refer to the https://openaccess.thecvf.com/content/CVPR2021/papers/Ge_Disentangled_Cycle_Consistency_for_Highly-Realistic_Virtual_Try-On_CVPR_2021_paper.pdf

**Strengths:**

1. The cycle-consistency strategy greatly enhances the model's ability to preserve clothing textures and structures, resulting in more realistic try-on outcomes.

2. The dual-network design enables various applications, such as clothing extraction and swapping between persons.

3. By utilizing clear frontal views for supervision, CycleVTON enhances adaptability to clothing deformations and variations in human poses, improving both supervision and robustness.

4. It demonstrates superior performance, as the model surpasses state-of-the-art (SOTA) methods on the VITON-HD benchmark, excelling in both qualitative and quantitative measures, and highlighting its effectiveness in virtual try-on.

**Weaknesses:**

1. Significant clothing deformations or angle changes lead to artifacts such as wrinkling or stretching, which limit the model's ability to accurately reproduce the original textures and patterns.

2.  It depends on the standardized front views for optimal performance, which may limit its effectiveness with certain types of clothing.

3. The dual-network cycle-consistent approach may raise computational costs and complexity, potentially affecting its use in real-time or resource-limited environments.

4. The cycle idea is very close to this paper, so the novelty needs to be clarified.
https://openaccess.thecvf.com/content/CVPR2021/papers/Ge_Disentangled_Cycle_Consistency_for_Highly-Realistic_Virtual_Try-On_CVPR_2021_paper.pdf

**Questions:**

Overall, CycleVTON represents an innovative approach to virtual try-on, balancing accuracy, flexibility, and robustness, with areas for improvement in handling complex clothing variations. But the cycle-consistent idea is not new, and please provide a detailed discussion on the paper: https://openaccess.thecvf.com/content/CVPR2021/papers/Ge_Disentangled_Cycle_Consistency_for_Highly-Realistic_Virtual_Try-On_CVPR_2021_paper.pdf

---

### Official Review · Reviewer_tsYe · 2024-11-04

**Soundness:** 3
**Presentation:** 3
**Contribution:** 2
**Rating:** 5
**Confidence:** 4

**Summary:**

This paper presents CycleVTON, a framework for virtual try-on. CycleVTON consists of two main components: a virtual try-on network that synthesizes human images given a specific clothing style, and a clothing extraction network that generates clothing templates in a unified space. Experiments are conducted on VTON-HD to evaluate the performance of the proposed approach.

**Strengths:**

The paper is well-written and easy to follow.
The proposed method outperforms existing baseline approaches on the VTON-HD dataset.

**Weaknesses:**

The idea of extracting human feature and clothing feature in a cycle framework is not novel, i.e., a similar work CycleVTON: A Cycle Mapping Framework for Parser-Free Virtual Try-On [Du et al., AAAI 2024]. The paper did not cite or discuss the CycleVTON paper [Du et al.]. A comparison against CycleVTON [Du et al.] should be provided since the ideas of the two papers are similar.

The technical contribution is limited. The virtual try-on part (ReferenceNet and denoising UNet) is borrowed from Animate Anyone [Hu et al.], and the main contribution is the idea of a clothing extraction network, which, however, has appeared in a previously published paper CycleVTON [Du et al.]. The technical contributions should highlight the differences against existing works.

No discussion on the failure cases (such as particular clothing types and poses). What are the limitations of the method?

**Questions:**

Can the CEN generalize to new datasets (such as DeepFashion)?
Does the CEN affect the generalization capability of the virtual try-on?

---

### Official Review · Reviewer_vBbk · 2024-11-04

**Soundness:** 2
**Presentation:** 3
**Contribution:** 2
**Rating:** 5
**Confidence:** 5

**Summary:**

This paper proposes CycleVTON, which is a virtual try-on framework that uses a cycle-consistent diffusion-based approach. It consists of two conjugated networks: a regular try-on network and a clothing extraction network. The clothing extraction network standardizes clothing into a front-facing format, allowing for alignment between generated and real images. This cycle-consistent optimization strategy enhances the retention of clothing textures and structures, ensuring realistic and accurate clothing generation. The conjugated network structure supports traditional virtual try-on as well as flexible clothing extraction and exchange between individuals. Experiments on VITON-HD demonstrate its effectiveness.

**Strengths:**

1. This paper proposes a new cycle-consistent diffusion-based framework for virtual try-on, named CycleVTON.

2. This paper introduces two complementary networks in CycleVTON. Alongside the conventional try-on network, we innovatively incorporate a clothing extraction network that extracts clothing from the human body and normalizes it to a frontal view.

3. This paper proposes a cycle-consistency optimization strategy, which aligns the generated dressed human with real images and the extracted clothing with its true frontal appearance.

**Weaknesses:**

1. In the proposed architecture, the CLOTHING EXTRACTION NETWORK differs only in its function, yet it does not fundamentally diverge from the ReferenceNet used in previous methods, such as IDM-VTON.

2. The proposed Cycle-Consistency Optimization Strategy is successful. However, there is a lack of validation for the effectiveness of its subprocesses and sub-losses, such as $L^_h$ and $L^_c$.

3. Some of the data in Table 1 seem to differ from those in the original paper, why?

4. Please provide the code to demonstrate reproducibility.

5. The entire process is overly engineered, lacking rigorous theoretical validation in this type of conference.

**Questions:**

Please refer to "Weaknesses."

---

### Note · Authors · 2024-12-11

I have read and agree with the venue's withdrawal policy on behalf of myself and my co-authors.